# Integral Use of Red Wine Pomace after Hydrostatic High Pressure: Application of Two Consecutive Cycles of Treatment

**DOI:** 10.3390/foods13010149

**Published:** 2024-01-01

**Authors:** Matilde D’Arrigo, Jonathan Delgado-Adámez, Javier Rocha-Pimienta, M. Esperanza Valdés-Sánchez, M. Rosario Ramírez-Bernabé

**Affiliations:** Technological Agri-Food Institute (INTAEX), Centro de Investigaciones Científicas y Tecnológicas de Extremadura (CICYTEX), Avda Adolfo Suárez s/n, 06071 Badajoz, Spain; matilde.darrigo@juntaex.es (M.D.); jonathan.delgado@juntaex.es (J.D.-A.); javier.rocha@juntaex.es (J.R.-P.); esperanza.valdes@juntaex.es (M.E.V.-S.)

**Keywords:** byproduct, integral red grape pomace, circular economy, hydrostatic high pressure

## Abstract

The influence of applying hydrostatic high pressure (HHP) to red grape pomace cv. Tempranillo was studied to obtain an ingredient rich in bioactive compounds for the manufacture of food products. Four treatments were investigated: (i) 600 MPa/1 s; (ii) 600 MPa/300 s, and other two treatments with 2 cycles of HHP: (iii) 2 cycles of 600 MPa/1 s; and (iv) 1 first cycle of 400 MPa/1 s and a second cycle 600 MPa/1 s. Treated pomace was stored at different temperatures (4 and 20 °C). The application of two consecutive cycles had no effect on the microorganisms’ inactivation compared to only one cycle. Immediately after HHP, the phenolic compounds content was maintained. However, HHP had no influence on the polyphenol oxidase enzyme (PPO), and so the phenolic compounds were significantly reduced during storage. Hence, the shelf-life of red grape pomace was significantly reduced at both temperatures, although phenolic compounds were better preserved under refrigeration than at room temperature.

## 1. Introduction

Wine production generates between 1.3 and 1.5 kg of waste per liter of wine, of which 75% is discarded, which is not very respectful of the environment and produces substantial losses of compounds with biological activity from these byproducts [1]. The principal byproduct of winemaking is grape pomace that produces 62% organic waste and consists mainly of seeds, skin, pulp, and stalk residues. In the case of red grape pomace (RGP), it is obtained after fermentation–maceration and the racking step (drawing off/devatting) [1,2]. Nowadays, grape pomace is utilized in conventional products of distillation, animal feeding, and biomass composting. In spite of this, grape pomace can be used in other alternative and innovative strategies to create products with an additional great value due to its chemical composition. Moreover, it can be applied in the manufacture of different foods such as meat, fish, bakery, and dairy products [3,4].

RGP is rich in polyphenols. Thus, it has antioxidant properties, which means numerous beneficial effects for health, such as those related against the incidence of cardiovascular and dermal diseases and various cancer types [4]. The phenolic profiles depend on the cultivar, the harvest, the geographical origin and/or cultivation conditions, as well as the extraction system used for the phenolic fraction [5,6]. Some studies have evaluated food fortification with RGP [4], such as yogurt (1–3%), pork sausages (0.5–1%), and cookies or a functional snack (6%). In some cases, lipid oxidation was retarded on meat products and yogurt. 

During the last few years, nonthermal processing methods, such as electric pulses, ultrasound, microwaves, supercritical fluids, and hydrostatic high pressure (HHP) have been studied for the valorization of grape pomace. Among them, HHP is one the most interesting technologies applied to recover thermolabile compounds from wine industry byproducts [7]. The application of HHP treatments on RGP produced significant increases in the extraction of total anthocyanin content [8,9]. Similarly, HHP showed promising results to recover bioactive compounds like phenolic compounds (including resveratrol) from the seeds and skins of cv. *Pinot Noir* [10,11]. Generally, bioactive compounds are extracted by HHP with the use of solvents and this process generates residues. Recently, however, new ways of applying this technology are being studied, since HHP would allow an “integral” use (skin, pulp, stalk residues, and seeds) of the winemaking byproduct without using solvents.

To obtain an ingredient from RGP rich in bioactive compounds, following the valorization treatment it should have a stable chemical and microbiological composition, with a minimal reduction in polyphenols, and low microbial counts (spoilage and pathogen microorganisms) to ensure the safety of the ingredient and a long shelf-life [12]. Concerning the chemical composition, González-Cebrino et al. [13] mentioned that the changes in bioactive compounds in fruit purées after HHP were closely connected to the remaining activity of the polyphenol oxidase (PPO) enzyme after processing, responsible of enzymatic browning reactions. The matter can be solved by a previous thermal blanching before HHP application, which achieves the inactivation of the enzyme [14]. Concerning the microbiological composition, bacterial spores can survive at HHP pressures above 1000 MPa, and these could be inactivated at 100 °C under conditions of pressurization [15]. However, the industrial units are restricted to using 600 MPa treatments since the pressure-transmitting fluid is water [16]. In consequence, the main goal has been to use HHP to make spores germinate and eradicate them in their vegetative state. A recent study conducted by our group [17] suggested the possibility of applying two cycles to increase the inactivation in the resistance forms of the spore microorganisms. The first cycle would activate spore forms; afterward, the vegetative cells from the spore forms would be inactivated during the second consecutive cycle. The application of multiple cycles of HHP, to eliminate foodborne pathogenic microorganisms, has been scarcely evaluated, especially for vegetable products. Timón et al. [18] applied two consecutive cycles to chicken burgers and found that treatments at 600 MPa/1 s (2 cycles) were more effective at reducing microbiological counts in comparison to 600 MPa/1 s. However, this effect on microorganisms was not observed at 400 MPa, when one or two cycles were applied. In addition, the effect of HHP on the whole grape pomace byproduct has been scarcely studied [19], and this technology would allow full use of the byproduct. The valorization of byproducts can be achieved through this green technology. The process does not generate any residuum and the whole byproduct can be valorized. HHP would allow for obtaining a grape pomace ingredient with safety and with long shelf-life, which could be used as an ingredient for the manufacture of other food products. However, the effect of storage conditions on the processed pomace has not been evaluated in depth, and this is important for a seasonal byproduct. The aim of the study was to attain chemical and microbiological stabilization in the RGP cv. *Tempranillo* by HHP (two consecutive cycles) and to know the effect of storage conditions on HHP-treated red wine pomace for its use as a potential bioactive ingredient in the manufacture of food products. 

## 2. Materials and Methods

### 2.1. Red Grape Pomace Sampling and Preparation

Ten kilograms of the RGP cv. *Tempranillo* were provided by Santa Marta de los Barros Coop. (Badajoz, Spain) in September 2020. The same procedure was carried out as described by Ramírez et al. [19]. RGP was vacuum-packaged in 1 kg plastic bags (OptiDureTM ODA7005 plastic bags, oxygen permeability: 10 cm^3^ m^−2^, 24 h^−1^ and 0% relative humidity, Cryovac, Madrid, Spain). Henkovac Proeco equipment (Henkovac International, Hertogenbosch, The Netherlands) was utilized for vacuum packaging (−0.8 bar). The packages were kept at −80 degrees Celsius until the experiment was executed. Then, frozen RGP was pulverized with a Thermomix TM5 (Thermomix-Vorwerk, Madrid, Spain) at maximum speed for around 3 min, and a mash like product was obtained. Subsequently, the milled pomace was homogenized and packaged in 50 g vacuum bags (with the identical composition as before). A total of 150 bags were prepared and stored at −80 °C until the application of the HHP treatment.

### 2.2. Treatment of Red Grape Pomace by HHP

The vacuum-packed RGP was prepared using semi-industrial equipment (6000/55, Hiperbaric, S.A., Burgos, Spain) with a 55 L volume vessel. The equipment is located in our institute. The initial temperature of the water during processing was 16 °C. Only the initial temperature of the water could be controlled because the vessel is a commercial unit without probes inside the container. Due to the low pH (4.05 ± 0.01) and aw (0.95 ± 0.01) of the RGP pomace, short holding times would be sufficient to reach the microbiological stabilization. Four alternative processes were implemented to create two pressure intensities, 400 and 600 MPa, according to our previous studies to avoid the residual growth of foodborne pathogens [17,18]. Four treatments were applied by combining different pressure levels and holding time according to the microbial effect found in previous studies concerning white wine pomace [19]: (i) HHP1: one single cycle of 600 MPa/1 s; (ii) HHP2: one single cycle of 600 MPa/300 s; (iii) 2 cycles (a): two cycles of 600 MPa/1 s; (iv) 2 cycles (b) one cycle of 400 MPa/1 s (first cycle) and other of 600 MPa/1 s (second cycle). When the two cycles were applied (2 cycles (a) and (b)), the first and the second cycles were separated for 3 h and 30 min. Thus, HHP1 and HHP2 consisted of one cycle while the last two treatments consisted of two separate cycles (2 cycles (a) and (b)).

The treated and the control vacuum-packaged samples were stored at −80 °C. In contrast, the other treated and control samples (three bags per treatment and day of storage) were stored for 1, 30, 90, 180, and 270 days in refrigerated storage (4 °C) or at room temperature (20 °C), both in the dark. To evaluate the sample stability, the color, pH, moisture, and a_w_ of the RGP after HHP were measured on each sampling day, while the other chemical analyses were performed on samples stored at −80 °C.

### 2.3. Determination of Moisture, Fiber, Protein, Fat, pH, and a_w_

The composition of the initial pomace was analyzed in 3 independent vacuum-packed bags, with untreated ground pomace. Moisture and protein analyses were determined following the methodology of AOAC; fat quantity was assessed using the Folch method [20] and fiber by the modified Southgate method [21], all of them in wet base (WB). Likewise, the pH was measured by a Crison pH meter (Crison, Barcelona, Spain), and water activity values using a Novasina Labmaster-aw meter (Novasina AG, Lachen, Switzerland) of the initial pomace were determined. 

### 2.4. Volatile Compounds

The principal volatile compounds and ethanol from the RGP were determined using 20 g of sample and 200 mL of water that were mixed in the initial step. This mixture was distilled in an oenological distiller (steam distillation system, DE 1626 GAB) to produce 200 mL of distillate. Analysis of volatile compounds in the distillates was performed using a gas chromatograph (Hewlett Packard 6890, Palo Alto, CA, USA) outfitted with FID. One microliter of each sample was injected into the gas chromatography equipment where the injector was kept at 250 °C and operated in split mode. Elution was realized in a capillary INNOWAX column (60 m × 0.32 mm i.d. × 0.5 µm). The oven temperature program was set to 50 °C for 5 min, followed by a linear ramp from 50 to 100 °C at 10 °C min^−1^, and finally to 220 °C at a rate of 30 °C min^−1^. The detection process was conducted using FID at a temperature of 250 °C. The identification was achieved by analyzing the retention times of standard compounds. Quantitative data were derived by interpolating relative peak areas in the calibration graphs created with known quantities of the analytes. 

### 2.5. Microbiology

Decimal dilutions of 10 g of RGP were aseptically made in sterile 0.1% (*w*/*w*) peptone water solution (Merck, Darmstadt, Germany). The total viable aerobic mesophilic counts were defined in plate count agar (PCA; Merck, Darmstadt, Germany) and incubated at 30 °C for 72 h according to ISO 4833-1:2013 [22]; molds and yeasts were counted using CG Agar Base (Merck, Darmstadt, Germany) with CGA Selective Supplement (Merck, Darmstadt, Germany), and incubated at 25 °C for 4–5 days according to ISO 21527-2:2008 [23] and *Enterobacteriaceae* (VRBG Agar, 37 °C, 24–48 h) according to ISO 21528-2:2017 [24]. Plates having 30–300 colonies were enumerated after incubation. The microbial counts were reported as the log of colony-forming units (CFU) per gram of sample (log CFU g^−1^).

### 2.6. Instrumental Color Measurement

CIELAB coordinates L* (0 = black, 100 = white), a* (−a* = greenness, +a* = redness), and b* (−b* = blueness, +b* = yellowness) were detected by a Konica Minolta CM-5 spectrophotometer, and a tray was used to perform the measurements (Konica Minolta, Tokyo, Japan). The global differences in color were evaluated by calculating the parameter ΔE: (ΔE* = ((L*_1_ − L*_2_)^2^ + (a*_1_ − a*_2_)^2^ + (b*_1_ − b*_2_)^2^)^0.5^).

ΔE processing contrasts color values of control-initial in red grape pomace against those tried with HHP. ΔE storage 1–30 d contrasts color values of initial RGP (day 1) with the pomace after treatment, at the same HHP conditions, for 30 days of storage. Thus for all of ΔE processing: ΔE storage 1–90 d, ΔE storage 1–180 d and, ΔE storage 1–270 d. 

### 2.7. Polyphenoloxidase (PPO) Enzyme Activity

Extraction and enzymatic activity of PPO were performed as described by Terefe et al. [25]. For kinetic modeling, absorbance was detected at 420 nm and 25 °C for 3 min with a Thermo Scientific Evolution 201 UV–Vis spectrophotometer (Fisher Scientific SL, Madrid, Spain). The results are presented as percent of activity relative to the control samples.

### 2.8. Total Content of Phenolic Compounds

The Folin–Ciocalteu reagent-based colorimetric assay, a method proposed and developed by Singleton and Rossi [26], with a slight modification , was used to quantify total phenolic content]. The absorbance was determined at 765 nm. using a Thermo-Evolution 201 spectrophotometer (MA, USA). A calibration curve was established with Gallic acid as the reference standard. Total phenolic content was expressed as Gallic acid equivalents per sample (wet base) (mg GAE 100 g^−1^).

### 2.9. Statistical Analysis of Data

The assay was executed in triplicate (three bags per group) and represented as mean ± SD. One-way ANOVA was used to analyze differences due to different treatments by applying the SPSS 21.0 statistical program (SPSS Inc., Chicago, IL, USA). Another one-way ANOVA was also used to evaluate variations in the samples during storage with identical HHP treatment and storage conditions. If the ANOVA method identified the existence of a statistically significant difference between means , these were compared with Tukey’s test (*p* < 0.05). Furthermore, a three-way (HHP, time, temperature) ANOVA interaction was applied to evaluate the effect of the factors studied. Pearson correlation coefficient (*r*) was calculated to assess bivariate correlations.

## 3. Results and Discussion

### 3.1. Proximate Analysis, pH, and aw of the Original RGP

The initial characterization of the red grape pomace is essential to know the possible applications of the byproduct. Results are expressed on a wet basis (Table 1) since the valorization treatment was applied on the wet byproduct; the valorized ingredient should also be distributed wet. In addition, some moisture content of food products is critical for microorganism inactivation after HHP. Generally, the proximate composition of pomace varies widely and this fact makes it difficult to compare results with reports in the literature. The content of moisture was analogous to data found by Jin et al. [3] for RGP from cultivars of *Petit Verdot*, *Merlot*, *Cabernet Franc*, and *Chambourcin* (50.7–58.1 g 100 g^−1^). The main component in RGP was the total fiber, a major component in RGP, which was higher than the results reported by Ramírez et al. [19], Teles et al. [27], and Xu et al. [28]. The protein content (4.5 ± 0.2%) was below those obtained by Deng et al. [29], Jin et al. [3], and Xu et al. [28]. These last authors confirm that samples of RGP after the fermentation and pressing showed a biomass of yeasts and bacteria resulting from high protein content. Seeds are the main source of fat in the pomace, and that factor determines the range of fat in the pomace. Fat content was similar to those reported by Theagarajan et al. [30], Sousa et al. [31], and Teles et al. [27]. Conversely, pH 4.0 in RGP allows for greater anthocyanin stability; pH > 4.5 alters different anthocyanin structures, which leads to viable fungal and bacterial growth. Thus, several factors, such as the maturity of the grapes at harvest and the conditions of the winemaking processes, explain the variability in the composition of the wine pomace reported in the literature [32]. The studies that analyzed the aw of grape pomace are few; however, Taşeri et al. [33] identified a similar amount to ours in a grape pomace from the *Hamburg Muscat* variety.

### 3.2. Effect of HHP Treatments of Volatile Compounds in RGP

A total of seven volatile compounds in the RGP after processing were isolated and quantified (Table 2). Ethanol was the most abundant compound, followed by acetaldehyde and methanol. Most of them have their origin in a fermentation process, since in the cellar, during storage of RGP, spontaneous anaerobic fermentation occurs, and the yeasts transform water-soluble carbohydrates into alcohols, esters, carboxylic acids, and aldehydes. However, methanol is not a compound from fermentation, but derived from grape pectin through the activity of the pectin methylesterase enzyme. It is well known that microbial activity on grape pomace increases the production of this enzyme [34]. 

As a general trend, HHP did not modify the values of volatile compounds. Ethanol presented similar values in the control and treated RGPs, so it was not modified after processing (*p* > 0.05). Since the pomaces analyzed came from red winemaking, their levels of ethanol and other alcohols are generally high, and their contents should be considered in the valorization of RGP as an ingredient for food production. Because of the high amount of alcohols, the RGP can modify the sensory attributes of the end-food; thus, as an attribute of its intense taste, the ingredient should be used at low doses. 

In contrast, only two compounds were modified after HHP while the other remained unchanged. Methanol, the second most abundant alcohol, significantly increased after the application of the two cycles of HHP. This is an unexpected result, and in this case, the application of two cycles would not be recommended, since methanol levels are generally limited in food products due to its toxicity. This change is difficult to explain since the changes in volatile compounds in RGP after HHP have not been previously evaluated.

In contrast, the levels of 2-phenylethanol were significantly decreased after the application of two cycles (a), while in the other treatments the levels of this compound showed intermediate values. Since the samples were analyzed immediately after processing, the changes in volatile compounds were more likely associated with physical–chemical changes in the pomace than to the modifications at the microbiological level. 

### 3.3. Microbiological Changes in High-Pressure-Treated RGP

When a three-way ANOVA was applied (Table 3), only mesophilic microbe count, molds, and yeasts were influenced by (1) HPP treatment, (2) storage temperature, and (3) storage time. *Enterobacteriaceae* counts were only affected by factors (1) and (3). Most factors showed significant interactions among them, especially P1 × P3, which was significant for the three microbial groups evaluated. 

After HHP (day 1), no significant differences in aerobic mesophilic count were observed compared to the control, while yeasts and molds and *Enterobacteriaceae* were significantly decreased later than processing (Table 4). HHP efficiency is influenced by extrinsic and intrinsic aspects: pressure intensity, process temperature, holding time, aw, microorganism species, and microorganism species growth phase [12]. A low minimal aw in the RGP could reduce the efficacy of the treatment on this product. However, the sublethal damage after HHP could enhance the impact of the treatments together with the days of storage [35]. In fact, at the end of storage in the control RGP, we observed an increase in the mesophilic aerobic microorganism count, 4.4 log CFU g^−1^ at room temperature and 1.5 log CFU g^−1^ for the refrigeration condition. Mold and yeast counts increased 3 log CFU g^−1^ at these temperatures. In contrast, such an increase was not found in the treated RGP that was kept for 270 days at refrigerated temperatures. The microbiological results for the RGP after HHP are within the microbiological criteria required by EC Regulation No. 2073/2005 and EC Regulation No. 852/2004 concerning the hygiene of food products. After using HHP, there were sublethal damages related to the incomplete loss of cytoplasmatic membrane function or injury to the outer membrane of the Gram-negative organisms. It does not lead to death of the cell, but is a potential survivor that may be selectively vulnerable to inhibitory mechanisms [35].

*Enterobacteriaceae* had an important diminution during the storage. This can be explained by the sensitivity of this group of microorganisms to acidic pH [32], which could be also enhanced during storage by acidification associated with the concentration of acids from the metabolism of the remaining microbial populations.

The application of two cycles of processing was evaluated to enhance the efficacy of the treatment for the inactivation of spores in RGP. The best-known mechanism for eliminating spores by HHP is achieved in two steps: first, pressures of 50 to 300 MPa are applied to germinate the spores and, then, thermal treatments and high pressures are carried out to kill the vegetative cells [15]. In the case of RGP, there were similar reductions in log CFU g^−1^ when HHP was applied on a single cycle (Control-HHP1: −2.4, Control-HHP1: −2.8), relative to a double cycle (Control-HHP3: −2.2, Control-HHP4: −2.5) for 270 days of preservation in refrigerated or temperature of the room, so that the utilization of two cycles would not offer any advantage. In contrast, Timón et al. reported that two cycles of 600 MPa/1 s were more effective than one single cycle in chicken burgers. Differences in the characteristics of the matrix (composition, pH, microbial population) could cause this opposite behavior [18]. The low pH and water activity of RGP can likely explain these differences relative to other products. Also, Rocha et al. [17] emphasized that the intensity of pressure and time have an important consequence of lethality against microorganisms. In conclusion, pressures higher than 300 MPa resulted in changes in cell membranes, which generated more injury with the increase in pressure and the time of exposition. 

### 3.4. Instrumental Color Measurement of High-Pressure-Treated RGP

When a three-way ANOVA was applied (Table 3), only CIE b* was affected by HHP. Lightness (CIE L*) and redness (CIE a*) were altered by the storage temperature. All parameters of color (CIE L, a*, b*) were significantly changed during the time of storage. Interactions between temperature and the time of storage were significant in the three parameters, and the interaction between HHP treatment and the storage time was important for CIE L* and a*.

In line with the previous statistical analysis, the CIE L* values were affected by HHP at any day of storage (Table 5); moreover, significant increases were found in the control at 4 °C and in all samples at 20 °C during storage. In addition, no differences were noted among HPP assays (*p* > 0.05) for the color parameter a*, although it was modified through time at both storage temperatures. There is no significant difference between the control and treated RGP for CIE b* after HHP on all sampling days, although b* values were slightly lower in the treated pomace (at day 1) than in the control. During storage, all groups (except HHP1 stored at 4 °C) showed a significant increase in b* after 270 days. The increment in CIE L* as CIE a* decreased during storage (270 days) and was more marked at 20 °C than at 4 °C. These results are in line with the interactions analysis for the factors of temperature and storage time (Table 3). 

Generally, pomace shows a dark red color, given the values of +a* and +b*, and reduced values of L* [36]. The color of the RGP from *Tempranillo* used in this study showed higher lightness and lower redness and yellowness than that reported by Xu et al. [28] on skin pomace of red grape (L*: 25.4, a*: 15.0, b*: 6.8). This pomace also contained branches that could partly explain the color differences. In addition, the composition and concentration of anthocyanins in RGP or the winemaking procedure, the type of waste, and many other factors could cause these differences [4].

According to our findings, Xu et al. [28] treated skins of freeze-dried red grape pomace from three varieties, *Merlot*, *Norton*, and *Petit Verdot*, at 600 MPa × 30 min. They verified the small influence of high hydrostatic pressure on the color of RGP. The cause of color retention after processing could be because the anthocyanins, the pigments responsible of red color, are stable under HHP treatment at moderate temperature [37]; this would agree with the maintenance of CIE a* values after HHP in our RGP. 

Regarding changes during storage, the increase in b* and decrease in a* would indicate a more intense yellowness and lower redness of the RGP, respectively. The degradation of anthocyanins during storage can explain this decrease in redness. In addition, the indirect oxidation of these compounds is also associated with the activity of the enzymes that produce enzymatic browning reactions and color changes [37].

Depending on ΔE values, the color difference specifies the degree of color change among processed and unprocessed RGP, which can be valued as 0–1.5 “not noticeable”, 0.5–1.5 “slightly noticeable”, 1.5–3.0 “noticeable”, 3.0–6.0 “well visible” and 6.0–12.0 “great” (6.0–12.0) [13]. ΔE values ranged from “not noticeable” to “noticeable” (0.3–2.3) changes (Appendix A). The parameter ΔE processing ranged between 0.3 and 0.8, so changes after HHP could be considered “not noticeable” when one cycle was applied. When two consecutive cycles were applied, changes were “slightly noticeable”.

Variations after 30 days of storage (ΔE storage 1–30 d) were “not noticeable” (1 cycle) or “slightly noticeable” (2 cycles) at both storage temperatures. Changes from 90 to 180 days in storage (ΔE 1–90 d, ΔE 1–180 d) were more marked at 20 °C than at 4 °C. At 20 °C, changes started to be “slightly noticeable”. At 180 and 270 days, changes were higher in the control than in the HHP-treated samples, which demonstrate the efficacy of the HHP treatment to stabilize the product during long-term storage. In addition, color changes were more intense at 20 °C than at 4 °C, so at 20 °C changes were “noticeable” at room temperature at the end of storage while samples stored at 4 °C showed “slightly noticeable” changes. Therefore, concerning color changes in RGP immediately after processing compared the storage conditions, changes in color after processing would be “not noticeable” (0–0.5) and “slightly noticeable” (0.5–0.8), while during storage, changes in color were “slightly noticeable” at refrigeration and “noticeable” at room temperature. Therefore, concerning color changes in RGP after processing and storage: (1) HHP would be recommended for long storage periods, but the temperature of the processed product should also be evaluated; (2) room temperature storage should be adequate only for short storage times (30–90 days), while for refrigeration, the color stability is higher than at 20 °C and allows for at least 270-day storage times.

Patras et al. [37] and Cao et al. [38] reported intense color modifications (ΔE) in strawberry pulp during storage (ΔE ≤ 3) and purées from blackberry (2.2–3.7) treated at 400, 500, and 600 MPa. Color changes in fruit products are rarely generated by HHP processing [39]. The instability of color in vegetables processed by HHP during their storage, however, is explained by the partial inactivation of enzymes and microorganisms after processing, which remained active during storage. These results are in line with the behavior of the RGP after processing, since slight color changes were found after the treatment, while changes during storage were more intense than after processing. As reported in these studies, ΔE during storage was higher than in ours, probably due to the greater stability of pomace compared with vegetable purées, the latter having high water or moisture content that facilitates chemical reactions or microbiological development (Appendix A).

### 3.5. Enzymatic Activity of Polyphenoloxidase (PPO) and Total Phenolic Compounds Content (PCC) of RGP Treated by High Hydrostatic Pressure

The three-way ANOVA (Table 3) demonstrated that the PPO enzyme activity was not altered by HHP. Otherwise, this activity was influenced by storage parameters (time and temperature). PCC was modified by three factors: HHP, time, and temperature of storage. Interactions were not significant for the PPO while PCC showed significant interactions between HHP × temperature, temperature × time, HHP × temperature × time. 

PPO activity was not modified after HHP (*p* > 0.05), although this enzyme had a small rise from the control to the treated RGP (Table 6). Neither the application of two consecutive treatments at the maximum pressure made any effect. Similarly, García-Parra et al. [40] studied the effects of HHP on plum and indicated that the treatment in some cases produces relative increases in the PPO activity, probably due to a greater interaction between the enzyme and substrate. 

At the end of storage, the PPO activity had decreased for all groups (control and HHP treatments), although reductions were stronger at room temperature. Regardless, it continued to be active after 270 days of storage at both temperatures, so large damage likely occurred in the phenolic compounds, which are the substrate of the enzyme, and would continue throughout the storage at room temperature. 

In contrast, the PCC was maintained after two cycles (600 MPa–1 s/600MPa–1 s), while the other HHP treatments reduced their content (74–92%) (Table 6). At day 30 and 270, HHP1 samples stored at 4 °C showed the lowest PCC. At 20 °C, no changes were observed between the control treatment and the RGP treated by HHP at 30, 90, and 270 days of storage, while at 180 days the control showed the lowest PCC. During storage at both temperatures, the control and HHP-treated samples showed large reductions in PCC. The reductions in PCC after 270 days of refrigeration were similar in the control and in the HHP-treated RGP. The PCC was preserved after 270 days of storage with respect to their initial content, ranging between 42 and 60%. Moreover, the reductions in PCC after 270 days of storage at 20 °C were very strong and only the 6% of the original content of the PCC in the control samples was preserved upon completion of storage, while in the treated samples by HHP, the percentage of retention ranged between 9 and 16% with respect to its initial content. Therefore, generally, after 9 months of storage, half of the phenolic compounds degraded at 4 °C while around 90% was lost at room temperature. The strong reduction in PCC during storage is contrary to making an ingredient from RGP with antioxidant or antimicrobial activity, which is the main objective of this study, since PCC is mainly responsible of the bioactivity in RGP [41]. Consequently, the red grape pomace samples had a shelf life of 90 days under the refrigerated storage conditions, in increasing order, corresponding to PCC: control < HHP1 < 2 cycles (a) and (b) < HPP2. PPO was effectively inactivated by a thermal blanching at 100 °C before HHP in white wine pomace [14], so that could also be a solution to inactivate the PPO of RGP.

Several studies have reported PCC (mg GAE 100g^−1^ DB) for untreated samples from red grape pomace of varieties such as *Cabernet Sauvignon* (1270–2670), *Merlot* (1830–2500), *Pinot noir* (1120–2140) [29], *Cabernet franc* (3610), *Petit Verdot* (6480), *Chambourcin* (1040) [3], *Tempranillo* (7762), and *Macabeu* (3093 ± 266) [6].

In the current study, HHP decreased or preserved the phenolic compounds in RGP immediately after the treatment, in contrast to previous results, which increased or maintained the extraction of PCC in the skins of *Dornfelder V. vinifera* ssp. byproducts [8,9], the ‘Summer Black’ grape from *V. vinifera × V. lambrusca* [42] and freeze-dried grape pomace from *Tempranillo*, *Petit Verdot*, and *Merlot* [28]. Also, Corrales et al. (2008, 2009) [8,9] and Sheng et al. [42] showed improvements in phenolic extraction from the grape skins throughout this treatment. This phenomenon is explained by changes in the structure of cellular matrices, especially in a matrix with dietary fiber content, leading the phenolic compounds extraction [40]. PCC was unchanged or decreased in HHP-treated *Merlot*, *Norton*, and *Traminette* skins from pomaces [28], in accordance with our study. 

The literature describes a close relationship between PPO activity and the stability of anthocyanins. Thus, the degradation of anthocyanins in processing berry products is the consequence of indirect oxidation of phenolic quinones by PPO and the peroxidase enzyme [37]. This would explain the similar actions found between PPO activity and color fluctuations in the analyzed RGP. Changes in instrumental color during storage could be explained by the activity of PPO, forming brown and colorless pigments. In fact, instrumental color parameters showed significant (*p* < 0.01) correlations (Pearson correlation coefficient) with PPO, which were negative for lightness and brownness (CIE L* r= −0.620; CIE b* r = −0.753) and positive for redness (CIE a* r = +0.327). Thus, the high activity of PPO produces reductions in brownness and lightness and increases in redness. Color modifications were more evident at room temperature than in refrigerated storage. However, the progress in the instrumental measurement of color was not as great as changes in PCC during storage. Probably, the important reduction in PCC was directly associated with the PPO activity in GP without blanching treatment, while the color changes are more related to the anthocyanin’s preservation, which is the secondary product of the reaction between the PPO and the phenolic compounds. This is an important concern for the valorization of pomace because these are the main bioactive portion of pomace, and they have antioxidant and antimicrobial activity, so their preservation is essential to maintain their biological activity [14]. Therefore, these authors concluded that a correct process of stabilization includes a thermal blanching of fresh pomace, grinding, vacuum packaging, and HHP (600 MPa/5 min), giving value to the integral grape pomace, which is abundant in phenolic compounds that the food industry could use as an ingredient.

There are no shelf-life studies for fresh pomace treated with HHP. Studies have been carried out, however, on dried pomace (skins and seeds) such as that by Tseng and Zhao [43] and Wang et al. [44] for red grape byproducts (*Pinot* and *Merlot*) maintained for up to 16 weeks at 15 ± 2 °C for 9 months. All of them achieved a dry product with a stable PPC, but these parameters decreased after 4 months. For example, in the study by Tseng and Zhao [43], the parameter values for *Pinot Noir* pomace decreased: 56% for PPC, 58% for anthocyanins, 36% for antiradical scavenge activity (ARS), and 35% for total flavonol content. Hence, HHP could be applied to obtain a shelf-stable product that retains PCC and lasts at least 9 months to 1 year, since RGP is a stationary product.

## 4. Conclusions

The application of two consecutive cycles of processing did not show any evidence for the inactivation of microorganisms when compared to the application of only one treatment cycle. The application of one cycle of 600 MPa (at holding times of 1 s or 300 s) allowed a sufficient microbial inactivation to reach a long shelf-life for the product (at least 9 month), which is important since this is a seasonal byproduct. Hydrostatic high pressure can be an appropriate system to process the red grape pomace to obtain a safe ingredient rich in phenolic compounds. In conclusion, HHP could be a suitable technology to valorize red grape pomace. HHP could make it possible to obtain an ingredient rich in fiber and with high levels of polyphenols. The treatment did not reduce the polyphenoloxidase enzyme and its activity, however, and therefore the phenolic compounds decreased significantly during storage. The remaining activity of the polyphenoloxidase enzyme during storage is another obstacle to avoid in order to preserve the content of bioactive compounds. This problem needs to be solved before having a commercial product with a long shelf-life. 

## Figures and Tables

**Table 1 foods-13-00149-t001:** Proximate analysis (g 100 g^−1^), pH, and aw of the RGP.

	Red Grape Pomace
Moisture	52.8 ± 0.3
Fiber	35.7 ± 1.8
Protein	4.5 ± 0.2
Fat	3.6 ± 0.5
pH	4.05 ± 0.01
aw	0.95 ± 0.01

**Table 2 foods-13-00149-t002:** Influence of HHP treatments on ethanol and volatile compounds in red grape pomace (concentration in mg kg^−1^).

	Control	HHP1	HHP2	2 Cycles (a)	2 Cycles (b)	*p*-Value
Aldehydes						
Acetaldehyde	381.2 ± 56.6	366.0 ± 34.7	356.2 ± 24.2	347.7 ± 32.4	291.9 ± 145.0	*0.330*
Alcohols						
Methanol	278.6 ± 169.8 ^3^	348.4 ± 67.9 ^2,3^	325.3 ± 33.2 ^2,3^	789.0 ± 237.0 ^1^	586.4 ± 217.3 ^1,2^	*0.000*
1-Propanol	22.0 ± 3.7	17.1 ± 8.6	35.6 ± 47.7	7.3 ± 11.4	0.0 ± 0.0	*0.104*
2-Methyl-1-propanol	3.4 ± 3.8	2.6 ± 1.9	3.9 ± 1.1	1.6 ± 1.9	2.1 ± 2.0	*0.427*
3-Methyl-1-butanol	133.1 ± 36.9	118.4 ± 19.0	139.1 ± 63.8	90.8 ± 19.3	84.6 ± 27.6	*0.066*
2-Phenylethanol	57.4 ± 4.3 ^1^	54.9 ± 2.3 ^1,2^	55.7 ± 5.8 ^1,2^	48.8 ± 2.2 ^2^	49.5 ± 6.2 ^1,2^	*0.010*
Ethanol	36,812.6 ± 8538.2	33,175.6 ± 5892.4	32,741.6 ± 2063.9	29,528.5 ± 4111.6	28,180.6 ± 8240.9	*0.193*

HHP1: 600 MPa/1 s; HHP2: 600 MPa/300 s; 2 cycles (a): 600 MPa/1 s (2 cycles); 2 cycles (b): 400 MPa/1 s (first cycle)/600 MPa/1 s (second cycle). Mean values annotated with different superscript numbers are significantly different among treatments, based on the Tukey test (*p* < 0.05).

**Table 3 foods-13-00149-t003:** Three-way ANOVA of the microbiological counts, CIE L*a*b color , polyphenol oxidase enzyme activity (PPO), and phenolic compounds content (PCC).

	Probability						
	P1	P2	P3	P1 × P2	P1 × P3	P2 × P3	P1 × P2 × P3
*Mesophilic*	*****	*****	*****	***	*****	*****	***
*Molds and Yeasts*	*****	*****	*****	*ns*	*****	*****	***
*Enterobacteriaceae*	*****	*ns*	*****	*ns*	*****	*ns*	*ns*
CIE L*	*ns*	*****	*****	*ns*	****	*****	*ns*
CIE a*	*ns*	*ns*	*****	*ns*	***	****	*ns*
CIE b*	***	*****	*****	*ns*	*ns*	*****	*ns*
PPO	*ns*	****	*****	*ns*	*ns*	*ns*	*ns*
PPC	****	*****	*****	*ns*	*****	*****	***

P1: *p*-value of HHP (hydrostatic high pressure); P2: *p*-value of storage temperature; P3: *p*-value of storage time. PPO: activity of Polyphenol oxidase; ns (nonsignificant differences). * *p* < 0.05, ** *p* < 0.01, *** *p* < 0.001.

**Table 4 foods-13-00149-t004:** Counts of microorganisms (log colony forming units, CFU g^−1^) in RGP treated with high pressure and stored at distinct temperatures.

	Refrigeration (4 °C)						Room (20 °C)					
	Control	HHP1	HHP2	2 Cycles (a)	2 Cycles (b)	*p-Value*	Control	HHP1	HHP2	2 Cycles (a)	2 Cycles (b)	*p-Value*
*Mesophilics*												
1 d	2.9 ± 0.9	2.7 ± 0.3 ab	2.6 ± 0.1	2.6 ± 0.4	2.5 ± 0.0 b	*0.763*	2.9 ± 0.9 c	2.7 ± 0.3 b	2.6 ± 0.1	2.6 ± 0.4 b	2.5 ± 0.0 b	*0.763*
30 d	2.9 ± 0.2 ^1^	2.2 ± 0.3 b ^2^	2.6 ± 0.2 ^1,2^	2.5 ± 0.2 ^1,2^	2.4 ± 0.1 b ^1,2^	*0.020*	4.8 ± 0.1 b ^1^	3.2 ± 0.2 b ^2^	3.4 ± 0.3 ^2^	3.3 ± 0.3 b ^2^	3.1 ± 0.2 ab ^2^	*0.000*
90 d	3.6 ± 0.7	2.9 ± 0.1 a	3.0 ± 0.5	2.3 ± 0.8	2.9 ± 0.1 a	*0.141*	5.3 ± 0.3 b	3.0 ± 0.0 b	4.6 ± 1.7	3.0 ± 0.0 b	5.8 ± 2.7 a	*0.118*
180 d	4.3 ± 0.2 ^1^	2.7 ± 0.1 ab ^2^	2.7 ± 0.2 ^2^	2.9 ± 0.1 ^2^	2.3 ± 0.0 b^2^	*0.000*	6.4 ± 0.4 a ^1^	3.3 ± 0.2 b ^2^	3.3 ± 0.3 ^2^	3.2 ± 0.2 b ^2^	2.9 ± 0.1 ab ^2^	*0.000*
270 d	4.4 ± 0.4 ^1^	2.8 ± 0.3 ab ^2^	2.4 ± 0.3 ^2^	2.5 ± 0.3 ^2^	2.2 ± 0.4 b^2^	*0.000*	7.3 ± 0.3 a ^1^	5.0 ± 0.5 a ^2^	4.7 ± 0.2 ^2,3^	4.1 ± 0.1 a ^3^	3.1 ± 0.1 ab ^4^	*0.000*
*p-storage*	*0.018*	*0.051*	*0.155*	*0.584*	*0.009*		*0.000*	*0.000*	*0.040*	*0.001*	*0.052*	
*Molds and yeasts*												
1 d	2.0 ± 0.9 b	<1	<1	<1	<1	*0.038*	2.0 ± 0.9 c	<1	<1 b	<1 b	<1	*0.038*
30 d	2.2 ± 0.2 b ^1^	<1 ^2^	<1 ^2^	<1 ^2^	<1 ^2^	*0.000*	3.7 ± 0.2 b^1^	<2 ^2^	<2 a ^2^	<2 a ^2^	<2 ^2^	*0.000*
90 d	2.6 ± 0.2 b ^1^	1.1 ± 0.2 ^2^	<1 ^2^	<1 ^2^	1.2 ± 0.3 ^2^	*0.000*	5.0 ± 0.2 a^1^	<2 ^2^	<2 a ^2^	<2 a ^2^	<2 ^2^	*0.000*
180 d	4.6 ± 0.2 a	1.3 ± 0.6 ^2^	1.4 ± 0.7 ^2^	<1 ^2^	<1 ^2^	*0.000*	4.5 ± 0.4 ab^1^	<1 ^2^	<1 b ^2^	<1 b ^2^	<1 ^2^	*0.000*
270 d	5.1 ± 0.1 a	<1 ^2^	<1 ^2^	<1 ^2^	<1 ^2^	*0.000*	5.0 ± 0.1 a^1^	<1 ^2^	1.4 ± 0.7 ab ^2^	1.3 ± 0.6 b ^2^	<1 ^2^	*0.000*
*p-storage*	*0.000*	*0.519*	*0.452*	*0.788*	*0.452*		*0.000*	*ns*	*0.005*	*0.001*	*ns*	
*Enterobacteriaceae*												
1 d	2.1 ± 0.7a ^1^	<1 ^2^	<1 ^2^	1.2 ± 0.3 ^2^	1.2 ± 0.3 ^2^	*0.021*	2.1 ± 0.7 a^1^	<1 ^2^	<1 ^2^	1.2 ± 0.3 ^2^	1.2 ± 0.3 ^2^	*0.021*
30 d	1.2 ± 0.3 b	<1	<1	<1	<1	*0.452*	<<1 b	<1	<1	<1	<1	*0.365*
90 d	<1 b	<1	<1	<1	<1	*0.751*	<1 b	<1	<1	<1	<1	*0.333*
180 d	<1 b	<1	<1	<1	<1	*0.421*	<1 b	<1	<1	<1	<1	*0.451*
270d	<1 b	<1	<1	<1	<1	*0.444*	1.6 ± 0.5 ab	<1	<1	<1	<1	*0.035*
*p-storage*	*0.012*	*0.888*	*0.828*	*0.452*	*0.452*		*0.019*	*0.888*	*0.901*	*0.452*	*0.452*	

HHP1: 600 MPa/1 s; HHP2: 600 MPa/300 s; 2 cycles (a): 600 MPa/1 s (2 cycles); 2 cycles (b): 400 MPa/1 s (first cycle)/600 MPa/1 s (second cycle). Data are expressed as mean ± SD (n = 3). Mean values with different superscript numbers in the same row or letters in the same column are significantly different among treatments or days of storage, respectively, as determined by the Tukey test (*p* < 0.05).

**Table 5 foods-13-00149-t005:** Instrumental color parameters for RGP treated at high pressure and stored at different temperatures.

	Refrigeration (4 °C)						Room (20 °C)					
	Control	HHP1	HHP2	2 Cycles (a)	2 Cycles (b)	*p-Value*	Control	HHP1	HHP2	2 Cycles (a)	2 Cycles (b)	*p-Value*
L*												
1 d	36.3 ± 0.1 b	36.4 ± 0.5	36.8 ± 0.2	36.9 ± 0.4	36.9 ± 0.1	*0.171*	36.3 ± 0.1 c	36.4 ± 0.5 b	36.8 ± 0.2 bc	36.9 ± 0.4 ab	36.9 ± 0.1 b	*0.171*
30 d	36.7 ± 0.4 ab	36.7 ± 0.3	36.7 ± 0.4	36.4 ± 0.5	36.7 ± 0.5	*0.913*	36.5 ± 0.3 c	36.3 ± 0.4 b	36.6 ± 0.4 c	36.3 ± 0.5 b	36.2 ± 0.3 c	*0.745*
90 d	36.8 ± 0.2 ab	36.8 ± 0.1	36.9 ± 0.0	37.1 ± 0.3	37.1 ± 0.2	*0.181*	37.2 ± 0.3 bc	37.3 ± 0.1 a	37.4 ± 0.2 ab	37.5 ± 0.2 a	37.7 ± 0.2 a	*0.096*
180 d	36.9 ± 0.5 ab	36.9 ± 0.5	36.7 ± 0.4	36.8 ± 0.3	37.1 ± 0.3	*0.863*	38.4 ± 0.7 a	37.7 ± 0.1 a	37.5 ± 0.3 a	37.6 ± 0.0 a	37.9 ± 0.3 a	*0.055*
270 d	37.4 ± 0.2 a	37.3 ± 0.2	37.0 ± 0.1	37.2 ± 0.1	37.1 ± 0.2	*0.180*	38.0 ± 0.4 ab	38.0 ± 0.1 a	37.9 ± 0.2 a	37.6 ± 0.6 a	37.6 ± 0.2 a	*0.373*
*p*	*0.031*	*0.136*	*0.702*	*0.146*	*0.449*		*0.001*	*0.000*	*0.001*	*0.005*	*0.000*	
a*												
1 d	4.0 ± 0.1 a	3.8 ± 0.5 ab	3.9 ± 0.1 ab	3.7 ± 0.2 ab	3.6 ± 0.2 b	*0.480*	4.0 ± 0.1 ab	3.8 ± 0.5 ab	3.9 ± 0.1 a	3.7 ± 0.2 ab	3.6 ± 0.2 b	*0.480*
30 d	4.1 ± 0.1 a	4.1 ± 0.1 a	4.1 ± 0.11 a	4.1 ± 0.2 a	4.2 ± 0.1 a	*0.928*	4.5 ± 0.2 a	4.1 ± 0.2 a	4.0 ± 0.2 a	4.1 ± 0.2 a	4.2 ± 0.1 a	*0.050*
90 d	4.1 ± 0.1 a	3.8 ± 0.1 ab	3.9 ± 0.1 ab	3.8 ± 0.1 ab	4.0 ± 0.3 ab	*0.270*	4.1 ± 0.3 ab ^1^	3.7 ± 0.2 ab ^2^	3.5 ± 0.1 bc ^2^	3.4 ± 0.2 b ^2^	3.6 ± 0.1 b ^2^	*0.004*
180 d	3.3 ± 0.2 b	3.5 ± 0.2 ab	3.4 ± 0.2 c	3.7 ± 0.3 ab	3.5 ± 0.2 b	*0.368*	3.8 ± 0.3 ab	3.6 ± 0.1 ab	3.8 ± 0.3 ab	3.7 ± 0.1 ab	3.6 ± 0.1 b	*0.539*
270 d	3.2 ± 0.2 b^2^	3.3 ± 0.1 b ^1,2^	3.5 ± 0.0 bc ^1,2^	3.3 ± 0.2 b ^1,2^	3.6 ± 0.1 b ^1^	*0.021*	3.6 ± 0.3 b	3.3 ± 0.1 b	3.4 ± 0.0 b	3.4 ± 0.4 b	3.5 ± 0.3 b	*0.782*
*p*	*0.000*	*0.019*	*0.001*	*0.016*	*0.004*		*0.033*	*0.048*	*0.004*	*0.025*	*0.003*	
b*												
1 d	0.4 ± 0.1 c	0.3 ± 0.1	0.3 ± 0.1 c	0.3 ± 0.0 c	0.3 ± 0.0 b	*0.809*	0.4 ± 0.1 b	0.3 ± 0.1 d	0.3 ± 0.1 d	0.3 ± 0.0 b	0.3 ± 0.0 c	*0.809*
30 d	0.4 ± 0.1 bc	0.4 ± 0.1	0.4 ± 0.1 bc	0.4 ± 0.1 bc	0.4 ± 0.1 b	*0.952*	0.6 ± 0.1 b	0.5 ± 0.1 cd	0.5 ± 0.0 c	0.6 ± 0.1 b	0.6 ± 0.11 bc	*0.561*
90 d	0.7 ± 0.1 ab	0.6 ± 0.2	0.6 ± 0.2 ab	0.7 ± 0.2 a	0.7 ± 0.2 a	*0.879*	0.8 ± 0.1 ab	0.7 ± 0.1 bc	0.8 ± 0.1 b	0.7 ± 0.1 b	0.8 ± 0.2 b	*0.529*
180 d	0.8 ± 0.1 a	0.6 ± 0.1	0.8 ± 0.2 a	0.7 ± 0.1 a	0.8 ± 0.1 a	*0.251*	1.4 ± 0.4 a	0.8 ± 0.1 b	1.1 ± 0.0 a	1.1 ± 0.1 ab	1.0 ± 0.1 ab	*0.073*
270 d	0.6 ± 0.2 abc	0.5 ± 0.1	0.7 ± 0.1 ab	0.6 ± 0.1 ab	0.8 ± 0.1 a	*0.142*	1.4 ± 0.1 a	1.1 ± 0.1 a	1.2 ± 0.0 a	1.5 ± 0.7 a	1.3 ± 0.3 a	*0.556*
*p*	*0.015*	*0.064*	*0.005*	*0.002*	*0.000*		*0.002*	*0.000*	*0.000*	*0.009*	*0.000*	

HHP1: 600 MPa/1 s; HHP2: 600 MPa/300 s; 2 cycles (a): 600 MPa/1 s (2 cycles); 2 cycles (b): 400 MPa/1 s (first cycle)/600 MPa/1 s (second cycle). Data are expressed as mean ± SD (*n* = 3). Mean values with different superscript numbers in the same row or letters in the same column are significantly different among treatments or days of storage, respectively, as determined by the Tukey test (*p* < 0.05).

**Table 6 foods-13-00149-t006:** Enzymatic activity of the polyphenol oxidase (PPO%) and the phenolic compounds content (PCC, mg GAE.100g^−1^) in RGP treated by high hydrostatic pressure and stored at different temperatures.

	Refrigeration (4 °C)						Room (20 °C)					
	Control	HHP1	HHP2	2 Cycles (a)	2 Cycles (b)	*p-*Value	Control	HHP1	HHP2	2 Cycles (a)	2 Cycles (b)	*p-*Value
PPO (%)												
1 d	100.0 ± 9.1	125.0 ± 21.7 a	145.8 ± 25.7 a	151.7 ± 35.5 a	139.2 ± 55.1 a	0.266	100 ± 9.1 a	125.0 ± 21.7 a	145.8 ± 25.7 a	151.7 ± 35.5 a	139.2 ± 55.1 a	*0.266*
90 d	65.8 ± 19.1	66.7 ± 17.6 ab	78.3 ± 23.1 b	58.3 ± 27.5 b	70.0 ± 13.2 ab	*0.825*	46.7 ± 24.7 b	49.2 ± 18.1 b	45.0 ± 13.2 b	45.0 ± 15 b	36.7 ± 5.8 b	*0.908*
180 d	45.0 ± 22.9	46.7 ± 47.5 ab	26.7 ± 2.9 b	36.7 ± 35.1 b	43.3 ± 17.6 b	*0.914*	1.7 ± 2.9 c ^2^	30.0 ± 5 b c ^1,2^	38.3 ± 15.3 b ^1^	26.7 ± 16.1 b ^1,2^	18.3 ± 17.6 b ^1,2^	*0.050*
270 d	45.0 ± 39.1	25.0 ± 31.2 b	27.5 ± 31.9 b	7.5 ± 13.0 b	22.5 ± 39.0 b	*0.726*	5.0 ± 8.7 c	8.3 ± 14.4 c	8.3 ± 7.6 b	6.7 ± 5.8 b	10.0 ± 10 b	*0.975*
*p*	*0.039*	*0.024*	*0.001*	*0.002*	*0.018*		*0.000*	*0.000*	*0.000*	*0.000*	*0.002*	
PCC												
1 d	633.3 ± 53.8 a ^1^	468. 8 ± 13 a ^3^	588.3 ± 20.6 a ^1,2^	626.3 ± 16.8 a ^1^	516.5 ± 69.4 a ^23^	*0.002*	633.3 ± 53.8 a ^1^	468.8 ± 13 a ^3^	588.3 ± 20.6 a ^1,2^	626.3 ± 16.8 a ^1^	516.5 ± 69.4 a ^23^	*0.002*
30 d	467.0 ± 31.8 b ^1^	217.2 ± 21.21 ^2^	476.2 ± 39.2 ab ^1^	461.3 ± 33 b ^1^	455.0 ± 83 a ^1^	*0.000*	403.0 ± 54.1 b	391.3 ± 86.3 a	341.4 ± 43.2 b	283.7 ± 47.3 b	330.8 ± 37.4 b	*0.141*
90 d	262.2 ± 11.7 c	297.4 ± 13.7 b	429.7 ± 98.7 b	334.3 ± 76 c	323.1 ± 25 b	*0.231*	241.6 ± 73.3 c	189.3 ± 10.6 b	204.7 ± 47.7 c	167.5 ± 63.4 c	148.7 ± 8.7 c	*0.246*
180 d	290.6 ± 5.6 c	334.6 ± 24.6 ab	275.0 ± 73.6 c	328.9 ± 8.9 c	330.7 ± 23.6 b	*0.392*	59.4 ± 27.8 d ^2^	169.8 ± 47.5 b ^1^	172.9 ± 11.9 c ^1^	150.7 ± 56.9 c ^1^	130.9 ± 15.1 c ^1,2^	*0.015*
270 d	264.5 ± 7.9 c ^1,2^	261.5 ± 15.6 b ^2^	266.1 ± 12.8 c ^1,2^	264.2 ± 34.6 c ^1,2^	311.2 ± 3.3 b ^1^	*0.034*	36.1 ± 8.7 d	52.4 ± 31 c	72.1 ± 15.2 d	63.0 ± 26.9 c	85.5 ± 5.6 c	*0.097*
*p*	*0.000*	*0.006*	*0.000*	*0.000*	*0.002*		*0.000*	*0.000*	*0.000*	*0.000*	*0.000*	

HHP1: 600 MPa/1 s; HHP2: 600 MPa/300 s; 2 cycles (a): 600 MPa/1 s (2 cycles); 2 cycles (b): 400 MPa/1 s (first cycle)/600 MPa/1 s (second cycle). Mean values with different superscript numbers in the same row or letters in the same column are significantly different among treatments or days of storage, respectively, as determined by the Tukey test (*p* < 0.05).

## Data Availability

The authors declare that the data supporting the results of this work are accessible within the article. The data presented in this study are available on request from the corresponding author.

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
