# Peer review of "Integral Use of Red Wine Pomace after Hydrostatic High Pressure: Application of Two Consecutive Cycles of Treatment"

_foods, 2024, doi:10.3390/foods13010149_

Round 1

Reviewer 1 Report

Comments and Suggestions for Authors

All comments and needed corrections are noted in the manuscripts' pdf file

Comments on the Quality of English Language

 Minor editing of English language required

Author Response

The answers are in the attached file

Reviewer 2 Report

Comments and Suggestions for Authors

The research analyzed how sequential hydrostatic high pressure (HHP) treatments influence the bioactive compounds in red grape pomace to be used as an ingredient for the manufacture of other food products.

It is an interesting study with a direct application. The manuscript is well written and technically sound. Following are my minor concerns which need to be addressed.

L32: Not clear. Please rewrite.

L40: Lipid oxidation in which products?

L43: What do you mean by “stabilization of red grape pomace”?

L51: Please elaborate more about “integral”.

L53: Please quantify the desired stability of pomace in terms of chemical composition and microbial perspective.

L68-70: Please establish the novelty while discussing the closely related studied and thereby finding out the research gap.

L88: What was the impact of -80C storage on enzyme activity in the sample?

L98: Please mention the pH and water activity.

 L102-106: The selection of the 4 treatments need justification. The authors claimed that the influence of second cylec was not explored in literature. However, it is imperative to separate the influence of first cycle from the total 2 cycles, so that the individual influence of second cycle can be identified.

L162: Why was the enzyme activity not expressed in terms of protein (enzyme) in the extract?

L210: Did the author check the PME activity in the pomace after HHP?

L254: When was the microbial count increased? Is it below the limit set?

-What was the upper limit of microbial tolerance in this sample? If it is 6 log cycle, then all the samples were microbially safe after 270 days. Please clarify this.

L276: Please quantify the log reduction values.

L317-318: Can you quantify the correlation between these color change and enzyme activity?

L340-344: Please emphasize on the threshold limit for total color change to decide this recommendation.

L364: This questions the selection of HHP conditions. It means that the sample was not enzymatically stable after HHP.

L371-372: What is the possible reason for this?

L387-389: Therefore, with respect to PCC, what was the shelf life of the pomace.

Table 6: Please check the enzyme activity after HHP per mg of protein extracted. Please note that enhancement in enzyme activity does not reflect enzyme activation.

L417-418: Quantitative correlation will be better here.

L450: Please mention whichever is performed in this study. Don’t extrapolate pls.

Please mention the temperature during HHP cycles.

Author Response

The answers are in the attached document

Round 2

Reviewer 1 Report

Comments and Suggestions for Authors

The quality of the manuscript is improved via revision